# Amaranth Seeds and Sprouts as Functional Ingredients for the Development of Dietary Fiber, Betalains, and Polyphenol-Enriched Minced Tilapia Meat Gels

**DOI:** 10.3390/molecules28010117

**Published:** 2022-12-23

**Authors:** Iza F. Pérez-Ramírez, Ana M. Sotelo-González, Gerardo López-Echevarría, Miguel A. Martínez-Maldonado

**Affiliations:** 1Facultad de Química, Universidad Autónoma de Querétaro, Querétaro 76010, Mexico; 2Instituto Politécnico Nacional, CICATA Unidad Querétaro, Querétaro 76090, Mexico; 3División de Gastronomía, Instituto Tecnológico Superior de Huichapan, Huichapan 42411, Mexico

**Keywords:** amaranth, *Amaranthus hypochondriacus*, dietary fiber, polyphenols, tilapia, sprouts

## Abstract

There is an increasing interest in the development of meat processed products enriched with antioxidant dietary fiber to augment the consumption of these health beneficial compounds. This study aimed to evaluate the nutritional, nutraceutical, and antioxidant potential, as well as the physicochemical properties of minced tilapia fillets (meat) gels with added amaranth seed or sprout flours (0%, 2%, 4%, 8%, and 10% *w*/*w*). Dietary fiber content was significantly increased with the addition of amaranth seed (1.25–1.75-fold) and sprout flours (1.99–3.21-fold). Tilapia gels with added 10% amaranth seed flour showed a high content of extractable dihydroxybenzoic acid and cinnamic acid, whereas the addition of 10% amaranth sprout flour provided a high and wide variety of bioactive compounds, mainly amaranthine and bound ferulic acid. The addition of amaranth seed and sprout flours increased hardness (1.01–1.73-fold) without affecting springiness, decreased luminosity (1.05–1.15-fold), and increased redness and yellowness. Therefore, amaranth seed and sprout flours could be used as functional ingredients for the development of fish products rich in bioactive compounds.

## 1. Introduction

Tilapia is a mild-flavored freshwater fish native to Africa but is currently farmed in over 135 countries around the world. The global market of tilapia increased from 3 million to over 6 million tons from 2010 to 2020 at a growth rate higher than 7%. Tilapia is one of the most consumed seafood worldwide and occupies fourth place in the USA market since it is a low-cost source of protein (about 14–19%) with a low content of fat (about 1.7–4.0%) [1,2].

Despite the high content of protein in fish and its derived processed products, one major disadvantage is the low content of dietary fiber, which is widely distributed in plant materials such as cereals, fruits, and vegetables. There is an increasing interest in the development of food products rich in dietary fiber to fulfill the recommendations of daily intake. Moreover, several dietary fiber sources are rich in antioxidant compounds, such as polyphenols, which provide added value, since antioxidant dietary fibers (ADFs) are considered human health promoters [3]. Currently, some consumers in some parts of the world are interested in convenient food products with high and complete nutritional value; therefore, the addition of ADFs to meat processed products is an interesting opportunity to satisfy their demands and to promote the consumption of processed foods with high nutrient value [4].

Interestingly, the addition of ADF sources not only increases the content of dietary fiber and antioxidant compounds of meat processed products but also modifies their physicochemical/techno-functional properties. In this regard, the addition of 6% cabbage powder or 25% oyster mushroom increased the hardness and springiness of mutton patties and chicken patties, respectively [5,6,7]. Conversely, the addition of 0.5–1% guava powder or 1–2% moringa flower negatively affected the texture of sheep and chicken nuggets [8,9]. Regarding fish-derived products, the addition of 2–4% grape pomace increased the hardness, springiness, and cohesiveness of anchovy mince without affecting its chewiness [10].

Therefore, the effect of ADF sources on the texture of meat processed products relies on the type and amount of dietary fiber, as well as the meat product. An interesting source of ADF is amaranth, which is a pseudo-cereal originally from America but is currently cultivated worldwide. Amaranth seeds are considered a low-cost source of dietary fiber, protein, and antioxidant compounds [11], which are increased during sprouting [12]. Therefore, this study aimed to evaluate amaranth seed and sprout flours as ADF sources to improve the nutritional and nutraceutical content, as well as the antioxidant and techno-functional properties, of minced tilapia fish meat (fillet) processed products.

## 2. Results

### 2.1. Evaluation of the Nutritional Composition of Minced Tilapia Meat Gels Enriched with Amaranth Seed or Sprout Flours

Amaranth sprouts showed a slight but significant (*p* < 0.05) decreased protein content as compared to amaranth seed flours (1.1-fold; Appendix A), but no significant changes were observed in the protein content of minced tilapia meat gels with added 2–10% amaranth seed or sprout flours (Table 1). On the other hand, amaranth fat content was slightly but significantly decreased in the sprout flour as compared to the seed flour (1.2-fold; Appendix A). Interestingly, the addition of amaranth seed or sprout flours significantly (*p* < 0.05) decreased the lipid content of minced tilapia meat gels from 1.2- to 2.2-fold (Table 1).

Amaranth sprout flour showed a 1.1-fold decreased carbohydrate content and a 3.08-fold increased dietary fiber content as compared to amaranth seed flour (Appendix A). Accordingly, the addition of 2–10% amaranth seed or sprout flours significantly increased the carbohydrate content of minced tilapia meat gels (2.1–4.9-fold and 2.1–4.5-fold, respectively), whereas dietary fiber was increased from 1.5- to 1.8-fold with 4–10% amaranth seed flour and from 2.0 to 3.2-fold with 2–10% amaranth sprout flour (Table 1).

### 2.2. Evaluation of the Polyphenol and Betalain Composition of Minced Tilapia Meat Gels Enriched with Amaranth Seed or Sprout Flours

The sprouting process significantly increased amaranth free polyphenol and flavonoid content (2.87- and 1.89-fold, respectively; Appendix A). The major extractable polyphenol identified in amaranth seed was vanillic acid, followed by dihydroxybenzoic acid, kaempferol rutinoside, and cinammic acid (Appendix A). Sprouting significantly (*p* < 0.05) reduced the content of some major extractable polyphenols of amaranth seeds, such as vanillic acid (1.65-fold), dihydroxybenzoic acid hexoside (1.68-fold), and cinnamic acid (2.34-fold) but augmented the content of kaempferol rutinoside (1.80-fold) and rutin (2.10-fold). This latter flavonoid was the second major extractable polyphenol identified in amaranth sprout flour, following vanillic acid. Interestingly, several polyphenols were only detected in amaranth sprout flours: hydroxybenzoic acid, caffeic acid, feruloylquinic acid, and ferulic acid.

Regarding the minced tilapia gels, the control sample (with added 0% amaranth) showed a high content of total polyphenols (150 mg/100 g); nevertheless, fish do not produce polyphenols (Table 2). The value obtained with this measurement is related to non-polyphenolic compounds, such as fish amino acids, which reduce the Folin–Ciocâlteu reagent. The addition of 2% and 4% amaranth seed flour did not modify the total polyphenol content in the minced tilapia meat gels, whereas the addition of 6% and 10% amaranth seed flour significantly (*p* < 0.05) decreased the free polyphenol content by 1.66- and 1.24-fold, respectively (Table 2). On the other hand, the addition of 2%, 4%, 6%, and 10% amaranth sprout flour did not modify the total free polyphenol content in minced tilapia meat gels (Table 2).

Interestingly, the minced tilapia meat gels with added 10% amaranth sprout flour showed the greatest content of free total flavonoids, which were increased by 1.64-fold as compared to the control sample (with added 0% amaranth), whereas the addition of 10% amaranth seed flour only increased the content of free flavonoids by 1.12-fold (Table 2), which is related to the higher content of free flavonoids in the sprout amaranth flour (Appendix A).

In this study, amaranthine and isoamaranthine were identified only in amaranth sprout flour (Appendix A). Minced tilapia meat gels with added 10% amaranth seed flour showed the highest content of dihydroxybenzoic acid and cinnamic acid; nevertheless, minced tilapia meat gels with added 10% amaranth sprout seed flour showed a greater variety of polyphenols, with a high content of bound ferulic acid (Table 3). Moreover, minced tilapia meat gels were enriched with amaranthine when 4–10% amaranth sprout flour was added, whereas isoamaranthine was not detected (Table 3) due to its low concentration levels in amaranth sprout flour (Appendix A).

### 2.3. Evaluation of the Total Antioxidant Capacity of Minced Tilapia Meat Gels Enriched with Amaranth Seed or Sprout Flours

The increased content of polyphenols and betalains in sprouted amaranth can be associated with an increased antioxidant capacity as observed in Appendix A with Q-ABTS and Q-DPPH radical-scavenging assays (12.3-fold as compared to amaranth seed flour). A minor effect was observed on Q-DPPH antioxidant capacity assay (1.6-fold; Appendix A). Regarding the addition of amaranth seed and sprout flours to minced tilapia meat gels, Q-ABTS and Q-DPPH antioxidant capacity was increased (Table 4, 1.7–3.9 and 4.4–5.9 fold as compared to the control, respectively), obtaining a higher antioxidant value when amaranth sprout flour was added.

### 2.4. Evaluation of the Techno-Functional Properties of Minced Tilapia Meat Gels Enriched with Amaranth Seed or Sprout Flours

The addition of 2–10% amaranth seed flour significantly (*p* < 0.05) increased the hardness and cohesiveness of minced tilapia meat gels, whereas no difference was found in springiness (Table 5). Regarding amaranth sprout flour, no clear trend was found, since the addition of 2% increased the hardness of minced tilapia meat gels without affecting their springiness and cohesiveness, whereas the addition of 6% increased the cohesiveness without affecting the other texture parameters.

On the other hand, color was significantly changed by the addition of amaranth seed or sprout flours (Δ*E* > 3; Table 6). The *L** value (lightness or darkness) and whiteness decreased with the addition of both amaranth seed and sprout flours at all concentration levels; nevertheless, all samples showed *L** values that indicated the presence of light (51–100). The *a** value (redness or greenness) was significantly (*p* < 0.05) increased with the addition of >6% amaranth seed flour but with >2% amaranth sprout flours due to its redness. Regarding *b** value (yellowness or blueness), this parameter was significantly increased with 10% amaranth seed flour and >2% amaranth sprout flour.

## 3. Discussion

In this study, amaranth sprouting decreased protein, carbohydrate, and fat content and increased dietary fiber content, leading to the development of minced tilapia meat gels rich in protein and dietary fiber and poor in lipid content. It has been reported that sprouting promotes protein hydrolysis due to an overexpression of endopeptidases. Nevertheless, controversial results have been reported regarding the effect of sprouting on the protein content of cereals [13]. Regarding carbohydrates, germination promotes the synthesis of α-amylase, β-amylase, and α-glucosidase enzymes, which degrade starch in simpler carbohydrates, leading to a decreased starch content in sprouted cereals as compared to the grains, thus increasing their digestibility [13].

Chauhan et al. [14] reported that germination slightly decreased amaranth’s carbohydrate content (1.03-fold) and slightly increased its dietary fiber content (1.36-fold). Similar results were reported by Perales-Sánchez et al. [12], who reported that sprouting decreased the already low-fat content of amaranth grains, which is related to an increased lipase and lipoxygenase activity during cereal germination [13].

Regarding the bioactive composition, Popoola [15] recently demonstrated that the extractable polyphenol content of *Amaranthus viridis* seeds increased after germination, which is related to an antioxidant defense mechanism against the increased production of reactive oxygen species generated after quiescent seeds initiate water imbibition. Popoola [15] reported ferulic acid as the major polyphenol in amaranth seed and sprout. Conversely, this hydroxycinnamic acid was not identified in amaranth seed flour in this study, but it was identified as both free and bound polyphenol in amaranth sprout flour. It is worth mentioning that, to the best of our knowledge, this is the first study that reports the bound polyphenol composition of germinated amaranth. On the other hand, betalains were only identified in amaranth sprout flour. Accordingly, Causin et al. [16] reported that betalains are synthesized during seed germination and seedling emergence in quinoa as part of its defense mechanisms.

The addition of amaranth seed and sprout flours did not proportionally increase the total polyphenol content in minced tilapia meat gels; nevertheless, this is related to the decreased content of fish amino acids when amaranth flours were added, since amino acids also react with Folin–Ciocâlteu reagent. Therefore, this UV/Vis spectrophotometric method is unreliable for assessing the polyphenolic composition of fish derived products since the UPLC–QToF-MS analysis demonstrated the enrichment with several polyphenols and betalains, which increased the antioxidant capacity of minced tilapia meat gels. Interestingly, amaranth sprout flour provided a higher concentration and variety of both hydrophilic and hydrophobic antioxidant compounds than amaranth seed flour.

Similar results were reported with the addition of other ADF sources to fish products. For instance, the addition of onion peel powder (1%, 2%, and 3%) increased the antioxidant capacity of fish sausages even at a very low concentration (1%) [17], whereas grape pomace dietary fiber (2%, 3%, and 4%) increased the antioxidant capacity of anchovy mince [10].

Lastly, regarding the techno-functional properties of minced tilapia meat gels, amaranth seeds and sprouts slightly affected the texture parameters, which may be related to its low lipid and high carbohydrate and protein content, since these latter macronutrients exert gelling effects, improving the stability and development of the protein network [4,18]. Similar results were reported by several authors, who added pseudo-cereals such as amaranth or quinoa flours (1.5–3% to different protein sources such as goat meat nuggets [19], beef burgers [20], and pork liver pâté [21], where only slight changes were observed on the TPA parameters.

Minced tilapia meat gels with added amaranth seed flour showed higher hardness values than the control samples and those with added amaranth sprout flour. It is noteworthy that hardness could be related to the interaction between amaranth and tilapia myofibrillar proteins which are partially unfolded during the heat-induced gelation process, exposing the sulfhydryl groups and internal nonpolar regions that interact to form aggregated structural proteins that further develop into a tridimensional protein matrix [22]. Nevertheless, the total protein content of both amaranth seed and sprout flours was similar; therefore, the higher content of dietary fiber of the amaranth sprout flour could negatively affect the formation of the tridimensional protein matrix, leading to a weaker network. Accordingly, García-Filleria and Tironi [23] reported that the addition of 1% and 2% amaranth protein isolate increased hardness in hake muscle, leading to the development of a fish restructured product with good texture attributes, whereas the addition carrageenan, konjac, and tragacanth as hydrocolloids rich in dietary fiber led to a lower hardness in fish ham and beef sausages [24,25].

On the other hand, the irregular trends observed in the TPA profile of minced tilapia meat gels with added amaranth flours could be associated with the polygonal shape of the starch granules of amaranth, as well as the release of amylose during the thermal process, which contributes to the formation of a protein–starch three-dimensional network. Nevertheless, starch swelling leads to a weaker network system, negatively affecting textural and rheological properties [26]. Lastly, regarding color parameters, similar results were reported by Felisberto et al. [27], who indicated that the addition of dietary fiber sources decreased luminosity of meat emulsions and by Verma et al. [19], who reported an increased redness in goat meat nuggets with added 1.5–3% amaranth flour.

## 4. Materials and Methods

### 4.1. Amaranth Seed and Sprout Flours

Amaranth (*Amaranthus hypochondriacus*) seeds were purchased from a local market in Querétaro, México. Seeds were previously disinfected with 0.1% *v*/*w* sodium hypochlorite (1:1.5 *w*/*v*) for 30 min at room temperature. For the germination process, seeds were soaked in water (1:1.5 *w*/*v*) for 1 h at room temperature (24–28 °C). Then, seeds were drained and washed with water. Hydrated seeds were placed in trays extended on a filter paper and covered. Germination conditions were set at 25 °C for 72 h in darkness. The filter paper was watered daily. Finally, sprouts were sun-dried for 24 h [28]. The germination process was carried out in triplicate. Amaranth seeds and sprouts were ground in a mill, and flours were stored at room temperature in darkness until analysis.

### 4.2. Minced Tilapia Meat Gels Enriched with Amaranth Seed or Sprout Flours

Fresh tilapia fillets (meat) were purchased in a local market in Querétaro, México. Tilapia fillets were minced and mixed with different concentrations of amaranth seed or sprout flours (0%, 2%, 4%, 8%, and 10% *w*/*w*) and 0.5% *w*/*w* sodium chloride to solubilize proteins. Then, the homogenized samples were stuffed into stainless-steel tubes and immersed in a water bath at 40 °C for 30 min, followed by a second immersion in a water bath at 90 °C for 20 min, and then cooled in iced water (4 °C) for 30 min. The minced tilapia meat gels were removed from the stainless-steel tubes and stored at 4 °C until analysis [29]. The control sample corresponded to the minced tilapia meat gels with added 0% amaranth seeds or sprouts. The concentrations of amaranth flours used in these studies were selected according to previous studies who added from up to 12% several dietary fiber sources [6,7,10]; nevertheless, we selected a maximum of 10% (*w*/*w*) since higher concentrations led to the formation of a fragile restructured product. Three independent batches of each treatment were prepared, and three samples were analyzed per batch. Representative photography of each treatment is included in Appendix A.

### 4.3. Nutritional Composition

The proximate analysis was determined following the official methods of analysis of the Association of Official Agricultural Chemists (AOAC): crude protein (method 920.87), crude fat (method 920.85), crude fiber (method 962.09), total ash (method 923.03), and moisture (method 925.10) [30]. The carbohydrate content was calculated with the following equation: $\%Carbohydrate=100-\left(\%Moisture+\%Crude\;protein+\%Crude\;fiber+\%Total\;ash+\%Crude\;fat\right)$. This analysis was performed in three independent experiments with three technical repetitions.

### 4.4. Polyphenols and Betalains Composition

For the polyphenol characterization, samples (0.5 g) were extracted with 20 mL of methanol/water (50:50 *v*/*v*) adjusted at pH 2 with hydrochloric acid (37% *v*/*v*) for 1 h at room temperature with constant stirring. Then, samples were centrifuged (1500× *g* for 10 min), and the supernatants were recovered. Residues were re-extracted with 20 mL of acetone/water (70:30 *v*/*v*) and centrifuged as previously described. Both supernatants were mixed and were considered the extractable polyphenol (EPP) fraction which was used for the determination of free polyphenols and flavonoids. On the other hand, both residues were mixed, dried (45 °C for 24 h), and considered the non-extractable polyphenol (NEPP) fraction, which was used for the determination of bound polyphenols [31]. This analysis was performed in three independent experiments with three technical repetitions.

For the betalain characterization, samples (0.5 g) were extracted with 5 mL of water for 2.5 h at room temperature with constant stirring. Then, samples were centrifuged (5000× *g* for 10 min at 4 °C), and the supernatants were recovered for the determination of betacyanins, betaxanthins, and betalamic acid [32]. This analysis was performed in three independent experiments with three technical repetitions.

#### 4.4.1. Free Polyphenols Content

Free polyphenols were determined in the EPP fraction. Samples (40 μL) were mixed with distilled water (10 μL), 1 N Folin–Ciocâlteu reagent (25 μL), and 20% sodium carbonate (125 μL) and were incubated for 30 min in darkness. Then, absorbances were measured at 765 nm. Results were expressed as μg of gallic acid equivalents/100 g fw [33].

#### 4.4.2. Free Flavonoids Content

Free flavonoids were determined in the EPP fraction. Samples (230 μL) were mixed 1 mg/mL of 2-aminoethyldiphenyl borate methanolic solution (20 μL). Then, absorbances were measured at 404 nm. Results were expressed as μg of rutin equivalents/100 g fw [34].

#### 4.4.3. Bound Polyphenols Content

Bound polyphenols were determined using alkaline hydrolysis in the NEPP fraction. Samples (0.3–0.5 g) were incubated with distilled water (12 mL) and 10 M sodium hydroxide (5 mL) for 16 h with constant stirring. Then, pH was adjusted to 2.0–3.0 with 6 M hydrochloric acid (37% *v*/*v*). Samples were centrifuged (2000× *g* for 10 min) and supernatants were recovered. Then, the residue was washed with distilled water (5 mL) and centrifuged as previously described. Both supernatants were mixed, and polyphenols were measured as previously described in Section 4.4. Results were expressed as mg of gallic acid equivalents/100 g fw [35].

#### 4.4.4. Betalain Content

Betalain pigments were determined in the betalain extract (Section 4.4). Samples (20 μL) were diluted with distilled water (180 μL). Then, absorbances were measured at 538, 480, and 430 nm for the estimation of betacyanin, betaxanthin, and betalamic acid content, respectively [25]. Betalain content was estimated using the following equation: concentration = (Abs × MW × DF)/(ϵ × L), where Abs is the absorbance, MW is the molecular weight of betacyanins (727 g/mol), betaxanthins (309 g/mol), or betalamic acid (212 g/mol), DF is the dilution factor, μ is the molar extinction coefficient of betacyanins (56,600 1/M·cm), betaxanthins (48,000 1/M·cm), or betalamic acid (24,000 1/M·cm), and L is the length.

#### 4.4.5. UPLC–QToF MSE Profile

An aliquot (1 mL) of the EPP, NEPP, and betalain fractions was evaporated to dryness at 35 °C under vacuum (Speedvac, Savant, Thermo Fisher Scientific, MA, USA), resuspended in 200 μL of methanol, and filtered with PVDF membrane syringe filters (0.45 μm, 13 mm). Then, samples were injected into a BEH Acquity C18 column (2.1 × 100 mm, 1.7 μm) at 35 °C in an ultraperformance liquid chromatograph (UPLC) coupled to a diode array detector (DAD) and a quadrupole/time-of-flight mass spectrometer (QToF MSE) with an electrospray ionization (ESI) interphase (Vion, Waters Co, MA, USA).

The mobile phase consisted of (A) water/formic acid 99.9:0.1 (*v*/*v*) and (B) acetonitrile/formic acid 99.9:0.1 (*v*/*v*) at 500 μL/min. The following gradient was used: initial conditions at 0% B, 0–15% B from 0 to 2.5 min, 15–21% B from 2.5 to 10 min, 21–90% B from 10 to 12 min, and 90–95% B from 12 to 13 min. Finally, conditions were returned to the initial 0% B from 13 to 15 min, which were maintained from 15 to 17 min to re-equilibrate the chromatographic column. Mass spectra were acquired in a mass range of 100–1200 Da in negative (ESI−) and positive (ESI+) ionization mode. MS conditions were set as follows: source temperature, 120 °C; desolvation gas (N_2_) temperature, 450 °C; desolvation gas flow; 800 L/h; cone gas flow, 50 L/h; capillary voltage, 2.0 kV (ESI−) and 3.5 kV (ESI+); cone voltage, 40 eV; low collision energy, 6 V; high collision energy, 15–45 V. Lock mass correction was carried out with leucine–enkephalin (50 pg/mL) at 10 μL/min every 3 min [36].

Data were acquired and processed in UNIFI software (Waters Co.). Peak identification was carried out by analysis of their exact mass (mass error < 5 ppm), isotope distribution, fragmentation pattern, and UV/Vis spectra. Calibration curves were constructed with commercial standards by triplicate, obtaining the regression coefficient, slope, and intercept for the quantification of the bioactive compounds. The limit of detection (LOD) and limit of quantification (LOQ) were quantified as three and 10 times the standard deviation of the intercept/slope, respectively (Appendix A). Representative high- and low-collision-energy mass spectra are included in Appendix A.

### 4.5. Antioxidant Capacity

For the QUENCHER-ABTS (Q-ABTS) assay, samples (10 mg) were mixed with 10 mL of an ABTS aqueous solution previously adjusted to an absorbance of 0.700 ± 0.002 at a wavelength of 734 nm. Samples were incubated for 30 min in darkness, and absorbances were measured at 734 nm. For the Q-DPPH assay, samples (10 mg) were mixed with 10 mL of a 150 mM DPPH methanolic solution previously adjusted to an absorbance of 0.700 ± 0.002 at a wavelength of 515 nm. Samples were incubated for 15 min in darkness and absorbances were measured at 515 nm. The percentage inhibition was plotted against time and the area under the curve was calculated. Results were expressed as μmol of Trolox equivalents/100 g fw [37].

### 4.6. Techno-Functional Properties

#### 4.6.1. Texture Profile Analysis (TPA)

Samples were compressed to 50% of their initial height (50 N load cell connected to the crosshead) at a compression rate of 50 mm/min using a 50 mm aluminum probe (P/50) (TA-XT plus Texture analyzer, Texture Technologies Co, Scarsdale, NY, USA). Hardness (peak force during the first compression cycle expressed in N), cohesiveness (area under the curve of the second compression cycle/area under the curve of the first compression cycle expressed as dimensionless quantity), and springiness (distance sample recovers after the first compression cycle expressed as mm) were recorded [30]. This analysis was performed in six independent experiments with three technical repetitions.

#### 4.6.2. Color Parameters

The spectral reflectance was determined using a HunterLab Mini Scan (MS/S-4000S, Hunter Associated Laboratory Inc., Reston, VA, USA) calibrated against white and black tiles. The CIE L, a, and b system was used to determine the color parameter values, and hue angle, chroma, and whiteness were calculated. This analysis was performed in six independent experiments with three technical repetitions.

### 4.7. Statistical Analysis

Results are shown as mean values ± standard deviation. Data normality and variance distribution were assessed with Kolmogorov–Smirnov’s and Levene’s tests. Then, data were analyzed by one-way analysis of variance (ANOVA) followed by the comparison of means by Tukey’s test (*p* < 0.05) using the JMP software (v14.0).

## 5. Conclusions

The results obtained in this study demonstrate that amaranth seeds and sprouts can be used to improve the nutritional and nutraceutical quality of tilapia restructured products without greatly affecting their techno-functional properties; however, color alteration must be considered in the development of a final food product. The elaboration of these fish meat gels with added antioxidant dietary fiber can be used for the development of hams, sausages, patties, and other processed fish products with a lower or null use of gelling additives. Further studies must be carried out to develop fish products using amaranth-enriched minced tilapia gels and to evaluate their sensory attributes and consumer’s preference. Moreover, a complete characterization of the final food product must be undertaken to guarantee its safety for consumers.

## Figures and Tables

**Table 1 molecules-28-00117-t001:** Nutritional composition of minced tilapia meat gels enriched with amaranth seed or sprout flours.

Minced Tilapia Meat Gel	Nutritional Composition ^1^					
	Protein	Lipids	Carbohydrates ^2^	Crude Fiber	Ash	Moisture
Control (0%)	15.15 ± 0.15a	1.44 ± 0.01a	1.96 ± 0.15e	1.35 ± 0.02e	3.41 ± 0.14a	78.03 ± 0.16a
+2% amaranth seed flour	14.99 ± 0.16a	1.00 ± 0.02b	4.14 ± 0.04d	1.69 ± 0.01e	2.91 ± 0.06b	76.95 ± 0.04ab
+4% amaranth seed flour	14.96 ± 0.14a	1.19 ± 0.05b	5.59 ± 0.55c	2.04 ± 0.01d	2.93 ± 0.04b	74.89 ± 0.73b
+6% amaranth seed flour	14.91 ± 0.21a	0.66 ± 0.02c	7.49 ± 0.23b	2.15 ± 0.06d	3.37 ± 0.01a	74.02 ± 0.07b
+10% amaranth seed flour	15.27 ± 0.22a	0.67 ± 0.00c	9.61 ± 0.55a	2.36 ± 0.06cd	3.18 ± 0.09a	71.29 ± 0.24c
+2% amaranth sprout flour	15.21 ± 0.11a	0.79 ± 0.02c	4.19 ± 0.09d	2.69 ± 0.01c	2.82 ± 0.02b	76.99 ± 0.01ab
+4% amaranth sprout flour	15.20 ± 0.23a	1.05 ± 0.04b	6.14 ± 1.05c	3.31 ± 0.01b	2.71 ± 0.04b	74.90 ± 0.82b
+6% amaranth sprout flour	14.72 ± 0.04a	1.01 ± 0.05b	7.62 ± 0.05b	3.88 ± 0.04ab	2.91 ± 0.05b	72.68 ± 0.11c
+10% amaranth sprout flour	15.67 ± 0.27a	1.11 ± 0.03b	8.90 ± 0.57a	4.34 ± 0.01a	3.56 ± 0.16a	71.41 ± 0.28c

Data are shown as mean ± standard deviation of three replicates. Different letters indicate significant (*p* < 0.05) differences between samples. Data are expressed as ^1^ % fw. ^2^ Calculated by difference. Fw: fresh weight.

**Table 2 molecules-28-00117-t002:** Free and bound polyphenol and betalain content of minced tilapia meat gels enriched with amaranth seed or sprout flours.

Minced Tilapia Meat Gel	Polyphenols				Betalains		
	Free Polyphenols ^1^	Free Flavonoids ^2^	Free Proantocyanidins ^3^	Bound Polyphenols ^1^	Betacyanins ^4^	Betaxanthins ^5^	Betalamic Acid
Control (0%)	150.08 ± 7.51a	51.77 ± 0.80c	3.30 ± 0.28c	261.64 ± 13.77a	0.13 ± 0.01f	0.04 ± 3 × 10^−3^e	0.08 ± 0.00f
+2% amaranth seed flour	146.78 ± 7.67a	59.58 ± 4.18bc	5.87 ± 0.40b	276.02 ± 12.56a	0.10 ± 0.00f	0.06 ± 2 × 10^−3^e	0.12 ± 0.01f
+4% amaranth seed flour	137.62 ± 4.93a	61.69 ± 5.08b	5.90 ± 0.42b	307.12 ± 25.04a	0.24 ± 0.01e	0.16 ± 9 × 10^−3^d	0.22 ± 0.01e
+6% amaranth seed flour	120.50 ± 4.96b	59.30 ± 1.17bc	5.59 ± 0.43b	306.85 ± 27.46a	0.26 ± 0.02e	0.21 ± 1 × 10^−2^c	0.23 ± 0.01e
+10% amaranth seed flour	121.17 ± 4.02b	57.85 ± 3.38bc	5.66 ± 0.03b	308.65 ± 20.12a	0.35 ± 0.01cd	0.22 ± 4 × 10^−3^c	0.37 ± 0.01d
+2% amaranth sprout flour	150.85 ± 4.68a	66.25 ± 3.37b	5.29 ± 0.59b	284.71 ± 24.25a	0.31 ± 0.01d	0.19 ± 3 × 10^−3^c	0.36 ± 0.01d
+4% amaranth sprout flour	138.75 ± 1.24a	65.52 ± 3.11b	7.26 ± 0.28a	302.09 ± 21.12a	0.40 ± 0.02bc	0.25 ± 2 × 10^−3^b	0.41 ± 0.00c
+6% amaranth sprout flour	137.88 ± 6.35a	64.09 ± 3.70b	7.17 ± 0.07a	311.52 ± 5.63a	0.42 ± 0.02b	0.28 ± 2 × 10^−2^b	0.46 ± 0.03b
+10% amaranth sprout flour	139.02 ± 3.84a	84.89 ± 3.04a	7.94 ± 0.18a	317.10 ± 8.60a	0.63 ± 0.04a	0.41 ± 9 × 10^−3^a	0.82 ± 0.00a

Data are shown as the mean ± standard deviation of three replicates. Different letters indicate significant (*p* < 0.05) differences between samples. Data are expressed as ^1^ mg of gallic acid equivalents/100 g fw, ^2^ mg of rutine equivalents/100 g fw, ^3^ μg of betacyanin equivalents/100 g fw, ^4^ μg of betaxanthin equivalents/100 g fw, ^5^ μg of betalamic acid/100 g fw. Fw: fresh weight.

**Table 3 molecules-28-00117-t003:** Free and bound polyphenol and betalain profile of minced tilapia meat gels enriched with amaranth seed or sprout flours.

Compound	Minced Tilapia Meat Gels								
	Control	+Amaranth Seed Flour				+Amaranth Sprout Flour			
	0%	2%	4%	6%	10%	2%	4%	6%	10%
**Free polyphenols**									
*Flavonols*									
Quercetin rutinoside (rutin) *	ND	ND	ND	ND	0.50 ± 0.02b	ND	ND	0.34 ± 0.00a	0.49 ± 0.02b
Kaempferol rutinoside	ND	ND	ND	0.37 ± 0.02a	0.64 ± 0.01c	ND	0.36 ± 0.01a	0.52 ± 0.03b	0.82 ± 0.06d
*Flavones*									
Apigenin dihexoside	ND	ND	ND	ND	ND	ND	ND	ND	ND
*Isoflavones*									
Daidzin hexoside	ND	ND	ND	ND	ND	ND	ND	ND	ND
Glycitin hexoside	ND	ND	ND	ND	ND	ND	ND	0.61 ± 0.05a	0.80 ± 0.02b
Genistin hexoside	ND	ND	ND	0.37 ± 0.03a	0.60 ± 0.16bc	ND	0.35 ± 0.01a	0.48 ± 0.02b	0.81 ± 0.15c
*Hydroxybenzoic acids*									
Hydroxybenzoic acid hexoside	ND	ND	ND	ND	1.09 ± 0.25a	ND	ND	1.12 ± 0.02a	1.52 ± 0.21a
Hydroxybenzoic acid *	ND	ND	ND	ND	ND	ND	ND	ND	ND
Dihydroxybenzoic acid hexoside	ND	0.84 ± 0.03a	1.21 ± 0.12b	3.40 ± 0.06e	6.75 ± 0.65f	1.07 ± 0.00b	1.62 ± 0.04c	1.97 ± 0.10c	2.90 ± 0.11d
Vanillic acid *	ND	ND	ND	1.07 ± 0.08a	2.11 ± 0.20b	ND	ND	1.12 ± 9.12a	2.13 ± 0.17b
*Hydroxycinnamic acids*									
Cinammic acid *	ND	ND	1.45 ± 0.04c	2.18 ± 0.19d	3.89 ± 0.24e	ND	ND	0.29 ± 0.03a	0.99 ± 0.10b
Ferulic acid hexoside	ND	ND	ND	ND	ND	ND	ND	ND	0.67 ± 0.05
Caffeic acid *	ND	ND	ND	ND	ND	ND	0.31 ± 0.02a	1.01 ± 0.07b	1.45 ± 0.11b
Feruloylquinic acid	ND	ND	ND	ND	ND	ND	ND	ND	ND
Ferulic acid*	ND	ND	ND	ND	ND	ND	0.09 ± 0.00a	0.34 ± 0.02b	0.71 ± 0.04c
**Bound polyphenols**									
*Hydroxycinnamic acids*									
Ferulic acid *	ND	ND	ND	ND	ND	10.33 ± 1.07a	13.11 ± 1.00a	15.54 ± 0.91b	18.47 ± 1.32c
**Betalains**									
*Betacyanins*									
Amaranthine	ND	ND	ND	ND	ND	ND	0.23 ± 0.01a	0.89 ± 0.04b	1.34 ± 0.08c
Isoamaranthine	ND	ND	ND	ND	ND	ND	ND	ND	ND

Data are shown as the mean ± standard deviation of three replicates. Results are expressed as μg/100 g fw. Different letters indicate significant (*p* < 0.05) differences between samples. * Identification confirmed with commercial standards. Fw: fresh weight.

**Table 4 molecules-28-00117-t004:** Total antioxidant capacity of minced tilapia meat gels enriched with amaranth seed or sprout flours.

Antioxidant Capacity	Minced Tilapia Meat Gels								
	Control	+Amaranth Seed Flour				+Amaranth Sprout Flour			
	0%	2%	4%	6%	10%	2%	4%	6%	10%
Q-ABTS assay	20.93 ± 1.61e	40.67 ± 0.34d	39.15 ± 2.40d	44.78 ± 2.70cd	36.09 ± 2.34d	57.95 ± 1.60bc	71.92 ± 4.03ab	74.36 ± 6.13a	80.79 ± 6.13a
Q-DPPH assay	16.86 ± 0.06d	74.65 ± 3.18c	83.04 ± 1.16bc	80.40 ± 7.56bc	85.17 ± 0.31b	94.57 ± 2.33a	97.93 ± 0.63a	98.66 ± 0.25a	97.04 ± 0.06a

Data are shown as the mean ± standard deviation of three replicates. Different letters indicate significant (*p* < 0.05) differences between samples. Data are expressed as μmol Trolox equivalents/100 g fw. Q: quencher; ABTS: 2,2′-azino-bis(3-ethylbenzo-thiazoline-6-sulfonic acid); DPPH: 2,2-diphenyl-1-picrylhydrazyl; fw: fresh weight; ND: not detected.

**Table 5 molecules-28-00117-t005:** Texture profile analysis parameters of minced tilapia meat gels enriched with amaranth seed or sprout flours.

Minced Tilapia Meat Gel	Texture Profile Analysis Parameters		
	Hardness ^1^	Springiness	Cohesiveness ^2^
Control (0%)	21.30 ± 2.44 a	0.92 ± 0.03 a	0.38 ± 0.01 b
+2% amaranth seed flour	28.25 ± 3.34 bc	0.93 ± 0.04 a	0.46 ± 0.05 c
+4% amaranth seed flour	32.95 ± 3.71 d	0.92 ± 0.03 a	0.42 ± 0.01 c
+6% amaranth seed flour	36.91 ± 3.61 d	0.95 ± 0.04 a	0.55 ± 0.01 d
+10% amaranth seed flour	36.75 ± 3.70 d	0.95 ± 0.03 a	0.40 ± 0.02 c
+2% amaranth sprout flour	26.55 ± 2.73 bc	0.94 ± 0.01 a	0.36 ± 0.01 b
+4% amaranth sprout flour	22.58 ± 2.55 ab	0.91 ± 0.01 a	0.21 ± 0.06 a
+6% amaranth sprout flour	21.57 ± 2.58 ab	0.93 ± 0.01 a	0.41 ± 0.02 c
+10% amaranth sprout flour	24.62 ± 1.28 ab	0.93 ± 0.02 a	0.23 ± 0.03 a

Data are shown as the mean ± standard deviation of six replicates. Different letters indicate significant (*p* < 0.05) differences between samples. Data are expressed as ^1^ N and ^2^ mm.

**Table 6 molecules-28-00117-t006:** Color parameters of minced tilapia meat gels enriched with amaranth seed or sprout flours.

Minced Tilapia Meat Gel	Color Parameters						
	*L**	*a**	*b**	Chroma	Hue Angle	Whiteness	Δ*E*
Control (0%)	71.45 ± 1.32a	−0.66 ± 0.51f	8.63 ± 0.89c	8.67 ± 0.87c	94.61 ± 3.48a	71.13 ± 1.31a	---
+2% amaranth seed flour	68.16 ± 1.61ab	−0.16 ± 0.32e	9.46 ± 0.58bc	9.47 ± 0.58bc	91.00 ± 2.00ab	67.86 ± 1.59b	3.54 ± 1.49c
+4% amaranth seed flour	68.04 ± 0.65ab	−0.11 ± 0.31e	9.81 ± 0.56bc	9.82 ± 0.55bc	90.72 ± 1.83b	67.74 ± 0.64b	3.71 ± 0.58c
+6% amaranth seed flour	65.41 ± 0.82b	0.17 ± 0.33d	10.46 ± 0.92b	10.47 ± 0.93b	89.20 ± 1.66b	65.11 ± 0.80bc	6.45 ± 0.65b
+10% amaranth seed flour	64.27 ± 0.87b	0.76 ± 0.43c	11.15 ± 1.04b	11.18 ± 1.06b	86.22 ± 1.79c	63.95 ± 0.86c	7.81 ± 0.89b
+2% amaranth sprout flour	68.35 ± 1.24ab	0.30 ± 0.39cd	11.70 ± 0.70b	11.71 ± 0.70b	88.61 ± 1.89b	67.98 ± 1.22b	4.59 ± 0.97c
+4% amaranth sprout flour	63.18 ± 2.70bc	0.52 ± 0.20c	12.99 ± 0.72b	13.00 ± 0.72b	87.69 ± 0.84bc	62.82 ± 2.65c	9.59 ± 2.09ab
+6% amaranth sprout flour	63.36 ± 1.52bc	1.76 ± 0.37b	16.25 ± 0.84a	16.34 ± 0.87a	83.87 ± 1.05d	62.87 ± 1.49c	11.45 ± 1.13a
+10% amaranth sprout flour	62.19 ± 0.59c	2.12 ± 0.29a	18.66 ± 0.81a	18.78 ± 0.81a	83.51 ± 0.85d	61.64 ± 0.57c	13.95 ± 0.72a

Data are shown as the mean ± standard deviation of six replicates. Different letters indicate significant (*p* < 0.05) differences between samples.

## Data Availability

The data presented in this study are available on request from the corresponding authors.

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
