# Peer review of "Amaranth Seeds and Sprouts as Functional Ingredients for the Development of Dietary Fiber, Betalains, and Polyphenol-Enriched Minced Tilapia Meat Gels"

_molecules, 2022, doi:10.3390/molecules28010117_

Round 1

Reviewer 1 Report

The manuscript is interesting scientific contributions to the knowledge of the amaranth seeds and sprouts as functional ingredients for the development of dietary fiber, betalains, and polyphenols enriched tilapia minced meat gels. The addition of antioxidant dietary fiber sources not only increases the content of dietary fiber and antioxidant compounds of meat processed products but also modifies their physicochemical/techno-functional properties. An interesting source of antioxidant dietary fiber is amaranth, which is a pseudo-cereal originally from America but is currently cultivated worldwide. In this regard, the aim of this study was to evaluate amaranth seed and sprout flours as antioxidant dietary fiber sources to improve the nutritional and nutraceutical content, as well as the antioxidant and techno-functional properties of tilapia fish minced meat processed products. The paper has high scientific level, the experiment is well designed, the discussion is consistent and the final conclusions are interesting. Therefore, the manuscript may be published in Molecules making major revision:

Suggestions for edition as well as some comments are the following:

Abstract

Please rewrite the abstract giving more information about the obtained results

Introduction

Please update the references. There are a lot of recent references about the use of antioxidant dietary fiber in foods. For instance:

https://doi.org/10.1016/j.tifs.2020.03.010

https://doi.org/10.3390/foods8080307

Line 240-241 why did you include these different concentrations of amaranth seed or sprout flours (0, 2, 4, 8, and 10% w/w)?

How many samples were analyzed for each batch? The batched were replicated on the time. Please include this information in the manuscript.

Please could you include some photos of the different batches

There are some mistakes through the text; for instance, in line 99, “The addition of 2 and 4% of amaranth seed flour did not modified..”. Please, check the English.

The discussion of results is very poor. Please try to improve using recent studies.

Conclusion

It´s missing. Please include some conclusions of your study.

I hope that my comments can improve the manuscript.

Author Response

Reviewer no.1

The manuscript is interesting scientific contributions to the knowledge of the amaranth seeds and sprouts as functional ingredients for the development of dietary fiber, betalains, and polyphenols enriched tilapia minced meat gels. The addition of antioxidant dietary fiber sources not only increases the content of dietary fiber and antioxidant compounds of meat processed products but also modifies their physicochemical/techno-functional properties. An interesting source of antioxidant dietary fiber is amaranth, which is a pseudo-cereal originally from America but is currently cultivated worldwide. In this regard, the aim of this study was to evaluate amaranth seed and sprout flours as antioxidant dietary fiber sources to improve the nutritional and nutraceutical content, as well as the antioxidant and techno-functional properties of tilapia fish minced meat processed products. The paper has high scientific level, the experiment is well designed, the discussion is consistent, and the final conclusions are interesting. Therefore, the manuscript may be published in Molecules making major revision:

Suggestions for edition as well as some comments are the following.

R. We appreciate the comment of the reviewer. We provide a point-by-point response to each comment made by the reviewer to improve the quality of our manuscript.

Abstract

Please rewrite the abstract giving more information about the obtained results

R. The abstract was modified as suggested (Lines 19-20 and 23-24).

Introduction

Please update the references. There are a lot of recent references about the use of antioxidant dietary fiber in foods. For instance:

https://doi.org/10.1016/j.tifs.2020.03.010

https://doi.org/10.3390/foods8080307

R. These references were included in the discussion section, but as suggested by the reviewer we included them in the introduction section to highlight the use of antioxidant dietary fiber sources in meat products (Lines 41-44 and 49-51).

Line 240-241 why did you include these different concentrations of amaranth seed or sprout flours (0, 2, 4, 8, and 10% w/w)?

R. These concentrations were selected according to previous studies described in the introduction and discussion section. Pérez-Jiménez et al. (2017) added 0, 2, 3, and 4% of grape pomace to anchovy mince, whereas Malav et al. (2015) and Mantihal et al. (2021), who added 0, 6, 9, and 12% of cabbage powder to mutton and chicken patties, respectively. Using these ranges as references, we carried out a preliminary study design, where we observed that concentrations >10% negatively affected the texture properties of tilapia mince leading to a fragile restructured product; therefore, the study design included a range of 0 to 10% of amaranth flour. We included a brief statement in the methodology section clarifying the selection of the range used in the experimental design (Lines 268-271).

Malav, O.P.; Sharma, B.D.; Kumar, R.R.; Talukder, S.; Ahmed, S.R.; Irshad, A. Antioxidant potential and quality characteristics of functional mutton patties incorporated with cabbage powder. Nutr. Food Sci. 2015, 45, 542-563.

Mantihal, S.; Azmi Hamsah, A.; Mohd Zaini, H.; Mantanjun, P.; Pindi, W. Quality characteristics of functional chicken patties incorporated with round cabbage powder. J. Food Process. Preserv. 2021, 45, e16099.

Solari-Godiño, A.; Pérez-Jiménez, J.; Saura-Calixto, F.; Borderías, A.J.; Moreno, H.M. Anchovy mince (Engraulis ringens) enriched with polyphenol-rich grape pomace dietary fibre: In vitro polyphenols bioaccessibility, antioxidant and physi-co-chemical properties. Food Res. Int. 2017, 102, 639-646.

How many samples were analyzed for each batch? The batched were replicated on the time. Please include this information in the manuscript.

R. Three independent batches of each treatment were prepared, and three samples were analyzed per batch This information was included in the methodology section (Lines 271-273).

Please could you include some photos of the different batches

R. A representative photography of each treatment was included in the supplementary material (Line 273; Figure 2S).

There are some mistakes through the text; for instance, in line 99, “The addition of 2 and 4% of amaranth seed flour did not modified.”. Please, check the English.

R. The English writing was revised throughout the manuscript.

The discussion of results is very poor. Please try to improve using recent studies.

R. The discussion was improved as suggested (Lines 216-237) and several references were added (Lines 481-498).

Conclusion

It´s missing. Please include some conclusions of your study.

R. A conclusion section was included at the end of the manuscript (Lines 396-406).

Reviewer 2 Report

From a purely biochemical viewpoint, I have no specific criticism; the work appears well performed with well described methods and is well written. 

From a nutritional viewpoint, at least at the European level, I do not see the interest of the approach that has been followed.  Why to mix in such a way tilapia minced “meat” gels and amaranth seeds and sprouts instead of proposing both ingredients separately to the consumers?  It is really difficult to understand for European people; if it is not the case for American consumers, this should definitively be explained by the authors…

From a technological viewpoint, my feeling is that to make from this association a consumable food, many additives should be included, providing therefore an “ultra-transformed” product, which is less and less accepted by consumers.  This item should also be discussed.  Furthermore, at least in Europe, based on my expertise, I am not convinced that such a product could be accepted by health and food safety authorities.  This aspect should be considered by the authors by distinguishing in their paper between a pure biochemical study and the development of a new nutraceutical ingredient.  The discussion should therefore be largely improved.

Finally, and much more important, since this work clearly aimed at developing the association of tilapia fish and a vegetable “… to improve the nutritional and nutraceutical content…”, I consider as mandatory an evaluation of the organoleptic properties of the product developed by a consumer panel.  This should be included or part of the paper should be rewritten to focus exclusively on the biochemical items.

A detail: being not English speaker, I wonder whether “meat” is appropriate for a product based on fish?

Author Response

Reviewer no.2

From a purely biochemical viewpoint, I have no specific criticism; the work appears well performed with well described methods and is well written.

R. We appreciate the comment of the reviewer. We provide a point-by-point response to each comment made by the reviewer to improve the quality of our manuscript.

From a nutritional viewpoint, at least at the European level, I do not see the interest of the approach that has been followed.  Why to mix in such a way tilapia minced “meat” gels and amaranth seeds and sprouts instead of proposing both ingredients separately to the consumers?  It is really difficult to understand for European people; if it is not the case for American consumers, this should definitively be explained by the authors…

R. Fish-based gels are used for the elaboration of several processed products such as ham, sausages, and patties. The proposed use of amaranth flour is to improve the physicochemical properties of the tilapia gel, leading to a lower or null use of additives such as starch which increases the glycemic index without providing any additional benefits like dietary fiber and antioxidant compounds found in amaranth seeds and sprouts. We included this perspective in the conclusion section (Lines 401-404).

From a technological viewpoint, my feeling is that to make from this association a consumable food, many additives should be included, providing therefore an “ultra-transformed” product, which is less and less accepted by consumers.  This item should also be discussed.  Furthermore, at least in Europe, based on my expertise, I am not convinced that such a product could be accepted by health and food safety authorities.  This aspect should be considered by the authors by distinguishing in their paper between a pure biochemical study and the development of a new nutraceutical ingredient.  The discussion should therefore be largely improved.

R. Most fish-derived processed products like ham, sausages, and patties include corn starch or other gelling additives to improve the texture attributes of the final product; in this study, we aim to evaluate if the addition of amaranth seed or sprout flour affects the technofunctional properties of tilapia minced gel, in addition to enriching the final product with antioxidant compounds and dietary fiber. We included this perspective in the conclusion section (Lines 401-404) and we further discussed the interaction between the components of amaranth (mainly starch and dietary fiber) and the protein of tilapia fish (Lines 216-237).

Finally, and much more important, since this work clearly aimed at developing the association of tilapia fish and a vegetable “… to improve the nutritional and nutraceutical content…”, I consider as mandatory an evaluation of the organoleptic properties of the product developed by a consumer panel.  This should be included, or part of the paper should be rewritten to focus exclusively on the biochemical items.

R. A sensory analysis was not included since tilapia minced gels are not a proper final food product, since further ingredients and flavorings are necessary to elaborate food products such as ham, sausage, and patties. We included a statement clarifying this point at the end of the manuscript as a perspective of the study (Lines 404-406).

A detail: being not English speaker, I wonder whether “meat” is appropriate for a product based on fish?

R. Meat is used to refer to the flesh of animals, including fish. We clarified the use of this term in the abstract, introduction, and methodology sections (Lines 18, 61, and 260).

Reviewer 3 Report

Detailed comments are included in the review file attached below.

Author Response

Reviewer no.3

A brief summary and general concept comments:

The aim of this study was to evaluate amaranth seed and sprout flours as antioxidant dietary fibers (ADF) sources to improve the nutritional and nutraceutical content, as well as the antioxidant and techno-functional properties of tilapia fish minced meat processed products. In the research, it was stated that dietary fiber content was significantly increased with the addition of amaranth seed and sprout flours to the tilapia minced products. Furthermore, it was proved that tilapia gels with 10% of amaranth seed flour showed a high content of extractable dihydroxybenzoic acid and cinnamic acid, while the addition of 10% of amaranth sprout flour provided a high and wide variety of bioactive compounds, mainly amaranthine and bound ferulic acid. Generally, it was concluded that amaranth sprouting decreased protein, carbohydrate and fat content and increased dietary fiber content, leading to the development of tilapia meat minced gels rich in protein and dietary and poor in lipid content. Finally, it was stated that amaranth seeds and sprouts can be used to improve the nutritional and nutraceutical quality of tilapia meat derived products without greatly affecting their techno- functional properties, but color alteration must be taken into consideration. In reviewer’s opinion, analyzing overall merit and rating interest to the readers and taking all the conclusions from the study into account, the results of this study may be of medium interest to readers. The results obtained and conclusions do not bring much novelty to the state of knowledge in this field. When it comes to the presentation of results, in the reviewer’s opinion the manuscript’s results are rather quite well presented, but the discussion of the results should have been more extensive, and surely it ought to be improved. What is also very crucial, subsection number 5 ‘Conclusions’ is missing in the manuscript. Furthermore, the validation of the polyphenols and betalains determination has not been carried out. This is a substantive objection and disadvantage of this article. Surely, the methods should be validated, and the method validation parameters should be presented and characterized in the manuscript, at least some of them.

R. We appreciate the comment of the reviewer. We provide a point-by-point response to each comment made by the reviewer to improve the quality of our manuscript.

The specific comments to the manuscript:

The section: ‘Discussion’ should be much more extensively written.

R.  The discussion was improved as suggested (Lines 216-237) and several references were added (Lines 481-498).

The section: ‘Materials and methods’ - the methods should have been validated and the method validation parameters should have been shown and discussed in the manuscript, at least some crucial validation parameters.

R. The validation parameters of the UPLC-ESI-QToF method were included in the supplementary material, including the range of the calibration curve, regression model, R2 coefficient, and the detection and quantification limits (Lines 354-358, Table 1S).

Line 200-201 - this sentence could be written in a different way, more precisely.

R. This sentence was modified as suggested by the reviewer (Lines 203-204).

In the manuscript the section no 5. Conclusion is missing. It is unthinkable to omit such an important part of the manuscript.

R. A conclusion section was included at the end of the manuscript (Lines 396-406), which includes a perspective for further studies.

Line 234-235 – Authors wrote: ‘Finally, sprouts were sun-dried for 24 h [21 with minor modifications]’ – It’s unclear what does this expression mean here? I can only guess that the methodological conditions taken from the given literature item have been changed. It doesn't look right written that way. Moreover, what modifications has been applied?

R. We apologize for the confusion. The full germination process is described in the methodology section (Lines 251-256). We eliminated the phrase ‘with minor modifications’ to avoid further confusion (Line 256).

Line 278-279 - Authors wrote: ‘[26 with minor modifications]’ - the same objection as above

R. We apologize for the confusion. The full processing conditions for the quantification of free polyphenols, flavonoids, and antioxidant capacity is described in the methodology section (Lines 303-306, 309-311, and 362-369). We eliminated the phrase ‘with minor modifications’ to avoid further confusion (Lines 306, 311, and 369).

Line 253 – Authors wrote: ‘The carbohydrate content was estimated by difference’ - it was written incomprehensibly.

R. We modified this sentence as follows: The carbohydrate content was calculated with the following equation: (Lines 279-281).

Line 316-318 - the sentence should be reformulated / corrected, also in terms of grammar.

R. We modified this sentence as suggested by the reviewer (Lines 341-346).

English level: is good and understandable, however the manuscript contains some linguistic errors. Among them, the following are as an example:

Lines 71, 107, 127, 142, 156, 171 – the same basic grammar error ‘are showed’ occurs in many places, also in the supplementary material

Spelling mistakes e.g.: Line 190, 199 – ‘defense’ - defence or ‘ Line 23 – ‘amaranthin’ – amaranthine

R. The English writing was revised throughout the manuscript.

Round 2

Reviewer 1 Report

Accept in present form

Author Response

We appreciate the comment of the reviewer.

Reviewer 2 Report

As a nutritional biochemist, once more, I have no real objection and I reaffirm that this work was well conducted and is clearly presented.

This work aims at developing new additives for ultra-transformed food.  Although less and less accepted in some parts of the world, i.e. Europe, this objective remains a credible goal, but it should be more precisely indicated in the paper.  For instance, the authors write in their revised version (l 41): “Nowadays, consumers are interested in convenient food products with high and complete nutritional value…  This is partially true, and only for part of the world and part of them.  A little more caution should improve the text, e.g.Some consumers… in some parts of the world …”.  L 44: the expression «wholesome foods » for such ingredients, not based on experimental evidences, should be avoided. 

Nevertheless, as an expert for food safety authorities in my European country, it is clear that the assertion “this study demonstrates that the amaranth seeds and sprouts can be used to improve the nutritional and nutraceutical quality of tilapia restructured products…” would not be accepted as such by European authorities to allow the launching of new products containing these ingredients.  Much more data should be added, e.g. the absence of pollutants, heavy metals, mycotoxins,…, toxicological evaluation,… to allow such a demonstration… 

Therefore, although the revised version is in progress and since situation may be different in Mexico than in Europe, I recommend to the authors to introduce more caution in their paper to increase its credibility at the international level.

Author Response

As a nutritional biochemist, once more, I have no real objection and I reaffirm that this work was well conducted and is clearly presented.

R. We appreciate the comment of the reviewer. We provide a point-by-point response to each comment made by the reviewer to improve the quality of our manuscript.

This work aims at developing new additives for ultra-transformed food.  Although less and less accepted in some parts of the world, i.e. Europe, this objective remains a credible goal, but it should be more precisely indicated in the paper.  For instance, the authors write in their revised version (l 41): “Nowadays, consumers are interested in convenient food products with high and complete nutritional value…”  This is partially true, and only for part of the world and part of them.  A little more caution should improve the text, e.g. “Some consumers… in some parts of the world …”.  L 44: the expression «wholesome foods » for such ingredients, not based on experimental evidences, should be avoided.

  1. We agree with the reviewer, we modified the sentence in the introduction section as recommended (Lines 41-45).

Nevertheless, as an expert for food safety authorities in my European country, it is clear that the assertion “this study demonstrates that the amaranth seeds and sprouts can be used to improve the nutritional and nutraceutical quality of tilapia restructured products…” would not be accepted as such by European authorities to allow the launching of new products containing these ingredients.  Much more data should be added, e.g. the absence of pollutants, heavy metals, mycotoxins,…, toxicological evaluation,… to allow such a demonstration… Therefore, although the revised version is in progress and since situation may be different in Mexico than in Europe, I recommend to the authors to introduce more caution in their paper to increase its credibility at the international level.

R. We agree with the reviewer, the food safety laws are different in Europe as compared to México. We modified the conclusion statement as recommended (Lines 407-408).

Reviewer 3 Report

In this version of the manuscript, the title of Table 1S should be changed because these validation parameters are not of chemical standards, but o the method used. This confuses the reader.  Moreover, there is no title in this Table 1S in line 418 of the manuscript (in the list of Supplementary Materials). Furthermore, in lines 356 -358 - in the abbreviation of LOQ the letter L is missing.  Other comments were well taken into account in the manuscript. Now only small changes should be made before publication

Author Response

In this version of the manuscript, the title of Table 1S should be changed because these validation parameters are not of chemical standards, but o the method used. This confuses the reader.  Moreover, there is no title in this Table 1S in line 418 of the manuscript (in the list of Supplementary Materials). Furthermore, in lines 356 -358 - in the abbreviation of LOQ the letter L is missing.  Other comments were well taken into account in the manuscript. Now only small changes should be made before publication

R. We appreciate the comment of the reviewer. We modified the title of Table S1 as suggested by the reviewer, which was included in the list of supplementary material (Lines 419-420). The abbreviation of LOQ was corrected (Line 358).